# Changing Veterinary Attitudes towards Delivering Biosecurity Advice to Beef Farmers

**DOI:** 10.3390/ani11071969

**Published:** 2021-06-30

**Authors:** Barnaby Edmund Howarth, Steven van Winden

**Affiliations:** Farm Animal Health and Production Group, Pathobiology and Population Sciences, Royal Veterinary College, Hawkshead Lane, Hatfield AL9 7TA, UK; svwinden@rvc.ac.uk

**Keywords:** biosecurity, beef cattle, veterinarians, competence

## Abstract

**Simple Summary:**

Biosecurity advice is an important way veterinarians can help farmers to reduce disease burdens on their farms. Many different factors are at play when delivering this advice, one being veterinary competence and their communication skills. This study looked at the private veterinary practitioners’ perceptions of their own competence to deliver biosecurity advice as part of a longitudinal biosecurity project spanning two years. Their responses were collected in the form of a telephone questionnaire. The results showed that as the project progressed the private veterinary practitioners felt more comfortable, better capable, and more consistent in giving their advice. In addition, they felt the uptake of their advice by the famers had improved throughout the study period. The mean average time spent delivering biosecurity advice increased and dropped subsequently, suggesting an initially more thorough process, and later a more efficient process. The results suggest development of the participating veterinarians following the conscious-competence learning model, showing a need to improve the knowledge and training of future generations of vets in the area of biosecurity with an increased focus on the importance of the veterinarian-farmer relationship in particular.

**Abstract:**

Biosecurity advice is an important way veterinarians can help farmers to reduce disease burdens on their farms. Many different factors are at play when delivering this advice, one being veterinary competence and their communication skills. This study looked at the private veterinary practitioners’ perceptions of their own competence to deliver biosecurity advice as part of a longitudinal biosecurity project. Their responses were collected in the form of a telephone questionnaire. The results showed significant increases in private veterinary practitioners’ responses to comfort (*p* = 0.022), capability (*p* = 0.002), and consistency (*p* = 0.006) as well as an increase of uptake of advice (*p* = 0.015) as the project progressed. The mean time spent delivering biosecurity advice increased and dropped subsequently, suggesting an initially more thorough and later on a more efficient process. The overall perceptions of the veterinarians of the study were also assessed. The results suggest development of the participating veterinarians following the conscious-competence learning model showing a need to improve the knowledge and training of future generations of private veterinary practitioners in the area of biosecurity with, in particular, an increased focus on the importance of the veterinarian–farmer relationship.

## 1. Introduction

Biosecurity is the practice aimed at keeping infectious diseases from populations and a key part of safe and efficient farming. It allows farmers to increase welfare of their animals and reduce production losses due to disease. Some farming sectors, such as the commercial pig industry, have imposed stringent biosecurity measures and shown the importance of keeping disease at bay in these large scale production systems and reducing reliance on antimicrobials [1,2,3]. In the cattle industry, however, there are marked differences. On dairy farms, private veterinary practitioners (PVP) are more involved on a regular basis with the management process of larger groups, ranging from fertility to mastitis and infectious disease control [4]. In commercial beef production more extensive systems are employed with less intense PVP involvement with their efforts mainly focusing on reproductive management.

Using PVPs as a means to spread awareness of the importance of disease prevention on farm enterprise has been deemed a key way for governments to spread policy regarding biosecurity [5,6]. Government agencies in the United Kingdom (UK), such as Department for Environment, Food & Rural Affairs (DEFRA) and the Animal and Plant Health Agency (APHA), use government appointed veterinarians to ensure PVP and farmers are following regulations, for example with auditing bovine tuberculosis (bTB) tests [7]. There is some difference of opinion as to whether this is the best way to increase farmer uptake of biosecurity measures, in particular related to non-notifiable, endemic diseases. Indeed, farmers may resent the use of government veterinarians to regulate [8]. These professionals may be perceived as being detached from the issue and not having the farmer’s best interests at heart, which would suggest that it is not the most effective way to increase uptake of biosecurity measures. PVPs, who have a relationship with the farmers, are far more likely to persuade farmers and increase preventative measures on farms [8].

PVPs, however, may face challenges when trying to give biosecurity advice to clients. Traditional farming methods and farmer-to-farmer communication can play an important role in forming opinions towards disease control. For example, the UK Randomised Badger Culling Trial (1998–2005) aimed to control the spread of *Mycobacterium bovis* from badger populations to cattle herds [9,10]. The relative success of the proactive culling program in high-risk areas showed a decrease in bTB cases in cattle and therefore demonstrated to farmers the success of this approach [9,10]. In high risk areas, such as Wales and the South West of England, a proactive badger cull gained support from local farmers as bTB is a constant battle for them [11]. The success of this culling trial has led to them deciding as a community that this is the way forward in control of this disease. A PVP may be faced with farmers who as a community want to adopt a single solution approach to biosecurity issues. Although it will help with building trust in the PVP and the advice given, it is not a realistic option in many endemic diseases as there are farm specific risks that need to be considered.

Because of their relationship with the farmer, a PVP may be able to play a key role in providing a greater variety of biosecurity advice. Farmers like to see results and the PVPs working closely with them can use this by explaining how more stringent biosecurity advice can bring benefits to the farmer. By showing these capabilities of the PVP the best possible outcomes for the farmer can be achieved [12,13]. 

This outcome is more likely when there is sufficient self-confidence in the PVPs’ own competence. In addition, an understanding of how to discuss disease risks with the farmer is essential. Being able to communicate concise, clear, and practical advice regarding these measures is important for uptake [14]. PVPs in the agricultural sector need to be given the training and support needed so that they understand the impact they can have. In return, they themselves will get a vested interest in the eradication process of disease [14]. Veterinary tact and technique with the individual farm clients is required, combined with well thought out biosecurity measures that can be tailored to individual circumstances. Compared to standard advice, this farm specific approach could provide a far more effective in increasing biosecurity on farms and therefore decreasing risk of disease [13].

The professional development of PVPs providing biosecurity advice could follow the learning cycle of the conscious-competence model [15,16]. This learning model is used in the development of clinical reasoning. At the first stage a naïve (unconscious incompetent) outlook is proposed, followed by a conscious incompetence and conscious competence and ultimately after more exposure unconscious competence would follow [15,16]. In order to gain insights to the perceived competence of PVPs advising on biosecurity, PVPs in a biosecurity project on beef suckler farms were interviewed in this longitudinal trial [17]. Aspects related to their perceptions as well as their performance (biosecurity score and visit time spent) were collected by structured questionnaire with the aim to explore the attitudes of PVPs working on the project and to suggest some answers to the challenges mentioned above.

## 2. Materials and Methods

### 2.1. Participant Recruitment

The data being analysed in this study was collected as part of a larger biosecurity project carried out by the Royal Veterinary College [17]. The biosecurity project focused on five endemic cattle diseases in the United Kingdom, and had 10 different veterinary practices as agents in the field; the practices recruited on average 12 beef suckler farms across Wales and the South West of England.

### 2.2. Biosecurity Scoring

An initial meeting to explain what the project was about, followed by an expert opinion workshop style of discussion to go through all aspects of risks associated with disease spread onto and within a cattle farm: biosecurity. The ten participating PVPs were briefed on previous work in which a farm-specific computer-based risk scoring tool had been developed. Evidence on generic risk factors for disease introduction on cattle farms was used to create generic risk factor categories: cattle purchasing, direct and indirect contact with other cattle, ruminants and other animals, use of shared equipment and types of visitors to the farm. These broad categories were divided into sub-factors to provide more detail. To elucidate risk factor weightings, the PVPs took part in two expert opinion workshops. During the workshops, PVPs were asked to allocate weights to reflect he relative importance of specific sub-factors, such that the total weight of all sub-factors with each broad risk factor category would be 100%. This participatory approach resulted in a reasonable agreement amongst the PVPs involved.

A semi-Delphi approach was used in order to achieve near-consensus, to achieve this the median scores were reported to the PVPs and discussed in detail, after which they scored the (sub)factors once again. The subsequent scores were used to construct a scoring tool in MS Excel™. The underlying algorithm generated an overall biosercurity score, with higher risk for disease introductions or spread and a lower scores for more biosecure units. The overall biosecurity score is the sum of factors contributing to the overall risk and the spreadsheet identifies the main risk contributor. This allowed farmers and PVPs to identify specific factors that could be targeted for change during the following year, and by altering these factors an aspirational score could be generated. Before the scoring tool was used on the farms, training was provided for participating PVPs, to familiarise them with the spreadsheet and to address any concerns.

The PVPs visited the farms annually in winter to take blood samples to identify which of the five endemic diseases are currently active on the farms. A risk assessment visit was then booked in spring where the farms were scored, the blood results were discussed and a plan of risk reduction was set out and agreed upon. The tool and the scores can be found in the supplementary materials as well as the technical results of the biosecurity project are reported previously [17].

### 2.3. Participant Perceptions

Alongside the biosecurity scores, specific data for this study was collected in the form of a questionnaire. The questions related to the individual PVP’s confidence and perceptions about delivering biosecurity advice at different stages of the study. The stages were, before the study commenced (before), one year into the study (Year 1), and two years into the study (Year 2). The answers to the questionnaire were collected by the author during phone interviews with the PVPs who took part in the study. Before the data collection took place, the questionnaire was trialed on a PVP not involved in the biosecurity project, ensuring the questions were worded appropriately.

Based on aspects of service management and the conscious-competence model a questionnaire was developed [15,16]. The questions relating to the PVPs’ perceptions that were repeated for each stage were marked on a scale of 1–10, 10 being fully able to identify with specific aspects of giving biosecurity advice. Aspects covered were: how comfortable, capable, and consistent the PVPs felt when giving advice. Also, the level of discretion the PVP felt when tailoring farm specific advice on biosecurity as well as the perceived uptake by the farmers of their biosecurity advice.

The PVPs were additionally asked to rate the overall impact on themselves as a result of the project participation: improved or increased knowledge, interest, involvement, likely role model or likelihood of charging for biosecurity advice. The project impact was graded using a five-point Likert scale (1 = ‘Strongly Disagree’ to 5 = ‘Strongly Agree’).

The amount of time spent on providing biosecurity advice by the PVP to the farmer was also recorded (in minutes) throughout the study. Time spent was captured in three measures: the shortest time, the mean time and the longest time taken to score the farms and give biosecurity advice. Whether the participating PVPs would be charging for biosecurity advice was captured in ‘No’, ‘Occasionally’, and ‘Always’ prior to the project as well as for the subsequent years.

Finally, themes were identified from the free text that the participating PVPs provided on the benefits and the disadvantages of taking part with the biosecurity project. Due to the limited number of participating PVPs, the thematic analysis of the free comments was performed manually. The questionnaire is attached in the Appendix A.

### 2.4. Statistical Analysis

The responses to the questionnaire were entered in a Microsoft Excel spreadsheet and transferred to SPSS (Version 26, IBM) for further statistical analysis. Normally distributed data were reported in mean and standard deviation (SD) and non-parametric data reported as median and interquartile rage (IQR). Spearman rank correlations were calculated between the responses in the different respective years. A Friedman test was used to evaluate the change in perception over time. A post hoc Wilcoxon Signed Rank Test was carried out to compare the respective years. A difference was considered significant if the *p*-value was lower than 0.05, after Bonferroni correction.

Evaluating the impact of the perceptions of the PVPs, as well as their time spent doing the biosecurity visits on the PVP’s specific biosecurity score, measured as mean score, the standard deviation (SD) of their scores and the coefficient of variation (COV). This allows exploring whether the biosecurity scores were getting more uniform (lower SD and COV) or whether PVPs felt more able to use the full breath of the scoring tool. These biosecurity measures were the dependent variable and tested by running a mixed linear model with PVP as mixed effect with year and the questionnaire responses as fixed effects. A multivariate approach was taken with a backwards stepwise approach, removing the least non-significant fixed effect variables until all variables were significant. Finally, the overall project participation was evaluated (improved or increased knowledge on, interest in, and involvement with biosecurity of cattle farms, likely to perceive themselves as a role model with respect to biosecurity for the farmer or likelihood, and charging for biosecurity advice) using the participant’s mean, SD and COV biosecurity score comparing ‘strongly agree’ responses versus lower Likert scores with a *t*-test.

## 3. Results

Of the ten biosecurity project PVPs, one veterinarian provided their perceptions before the study commenced and one year into the study, but not for the second year. We received complete data from eight participants, each from a different practice, giving a response rate of 80 percent. The eight PVPs represented 90 farms visits where biosecurity was scored and advice was given. The mean scores on these biosecurity visits was 150 (SD = 42) and it took them on mean 83 min (SD = 42). The shortest reported visits lasted on mean 56 (SD = 31) minutes and the longest 119 (SD = 54) minutes in duration. The mean time spend on farm scoring for biosecurity and giving advice on project farms on the two rounds of the project compared to their initial reported time spent giving biosecurity advice. The time spent on giving biosecurity advice went from 21 +/− 7 (mean +/− SD) minutes before the biosecurity project, to 93 +/− 42 min in year one (*p* < 0.001) and 69 +/− 39 min (*p* = 0.017) in year two of the project.

### 3.1. Participant Perceptions

The overall reported perceptions of the participating PVPs before (baseline), year one and year two reported values for ‘Comfortable’, ‘Capable’, ‘Consistent’, ‘Discretion’ and the level of advice ‘Uptake’ was (median, IQR): 8 (7.25–9), 8 (7.125–8), 8.25 (7.25–9), 8 (6.25–8.5), and 5 (5–7) respectively. The perceptions of the PVPs over the span of the biosecurity project are presented in Table 1 below. In all bar the level of discretion, the perception of these PVPs was graded higher as the project progressed (*p* < 0.05). 

The participating veterinarians reported charging for biosecurity advice prior to the project as ‘Occasionally’, (IQR: No-Occasionally), this was moved towards ‘Always’ (IQR: Always-Always). This shift was significant when evaluating this with the Wilcoxon Signed Ranks test (*p* = 0.030). Although the sense of being a role model for biosecurity moved from ‘Agree’ (IQR: Agree-Strongly agree) to ‘Strongly agree’ (IQR: Agree-Strongly agree), this was not significant (*p* = 0.102). The perceived role model prior to the project had a significant correlation with the level of comfort in giving biosecurity advice, r = 0.735 (*p* = 0.024), and again at Year 2 with the post role model feel: r = 0.716 (*p* = 0.046). There was, however, no correlation with comfortability of giving biosecurity advice during Year 1 of the project.

There were a few correlations (Spearman Rank) identified, which were not the same throughout the progress of the project. Before the project, ‘Consistency’ of the advice and the ‘Capability’ of the PVPs to give the advice showed a positive correlation coefficient of (r = 0.761) which was significant (*p* = 0.017). In year one, there was a positive correlation between the ‘Uptake’ of the advice by the farmer and the ‘Comfort’ of the PVPs delivering the advice (r = 0.772, *p* = 0.015). Furthermore, ‘Capability’ and ‘Comfort’ of the PVPs also showed a moderate positive correlation with (r = 0.670, *p* = 0.048). Finally, ‘Capability’ delivering information and ‘Consistency’ of advice revealed a positive correlation, (r = 0.767, *p* = 0.016). For year two, no significant correlations were found amongst the reported parameters.

### 3.2. Biosecurity Score

The overall mean biosecurity score of years 1 and 2 was 146.4, with a standard deviation of 86.9 and a coefficient of variation of 0.568. The linear mixed effects models revealed that for the mean biosecurity score, the second year had an mean 34.5 lower score and longer visits resulted in higher scores (*p* = 0.001). In addition, the mean time spent on the biosecurity consult visit resulted in a higher score (0.7/minute, *p* = 0.007), however, when combined in the multivariate approach, year was the only variable remaining in the model. The standard deviation of the biosecurity score was lower in PVPs reporting feeling more comfortable in giving biosecurity advice (−11.7 per score point, *p* = 0.042). Taking into account both the standard deviation and the mean, the coefficient of variation (SD/mean) increased with 0.128 per score of sense of capability (*p* = 0.003) and 0.059 per score of sense of discretion (0.024). The multivariate approach was left with capability of being the only significant parameter in the model.

The mean, SD and COV biosecurity score in year 1 was 163.4, 91.5, and 0.540 and in year 2129.4, 82.4, and 0.596 respectively. The mean score changed significantly (*p* = 0.001), the other measures of the biosecurity score did not (*p* = 0.131 and *p* = 0.100 respectively). Comparing the responses of the overall project participation (improved or increased knowledge, interest, involvement, likely to perceive themselves as a role model with respect to biosecurity for the farmer or likelihood of charging for biosecurity advice), responses of ‘strongly agree’ versus not, there are a few significant differences in the biosecurity scores. The mean score in year 1 was higher in PVPs that felt strongly that they had an increased interest in biosecurity: 177.1 vs. 135.9 (*p* = 0.037) and this was also the case in year 2: 147.5 vs. 93.1 (*p* = 0.019). Similarly, the SD and COV of their scores was higher in both years: SD year 1: 109.8 vs. 54.8 (*p* = 0.012), SD year 2: 102.7 vs. 41.6 (*p* = 0.023), COV year 1: 0.608 vs. 0.402 (*p* = 0.030) and COV year 2: 0.669 vs. 0.450 (*p* = 0.052). For the other overall project scores only improved knowledge had higher mean biosecurity scores in year 2 in PVPs that strongly agreed and the ones who did not: 165.1 vs. 111.5 (*p* = 0.045). 

The main themes on the benefits in taking part with the biosecurity project are an improved level of awareness, knowledge and interaction in both PVPs and farmers, testing for disease allowed opening up the discussion. The involvement in the biosecurity project allowed identification of disease presence on the farm and through that helped the case for vaccine sales or disease eradication. More specifically, five of the participating PVPs reported that they “increased their involvement on farm” and “farmers improved understanding of risks”. However, one PVP mentioned that “uptake wasn’t great” and three other PVPs mentioned they felt they had to “hassle farmers”, suggesting that individual experiences did vary. Overall, there were fewer disadvantages reported than advantages, and they mainly pivoted around the feeling of the need to persuade the farmers to take part.

## 4. Discussion

This study describes the changes in perceptions of PVPs that have taken part in a biosecurity project where they scored the level of biosecurity and gave biosecurity advice on beef cattle farms. The success of the use of the actual tool supporting the advice has been reported earlier [17] and has been echoed by other research groups as well [18]. The use of a structured questionnaire or tool to assess the level of biosecurity reduces observation bias. There are inevitable biases in the perception questionnaire described in this paper. Recall, conforming, and reporting bias during the collection of the PVPs’ perception, which necessitate caution with interpreting the results.

The current findings show an association between the training during the project and their ability to deliver advice effectively and eloquently: over the duration of the longitudinal biosecurity project, the time spent initially increased, dropping again the next year. The initial increase suggests that more thorough visits were taking place and more comprehensive advice was being given later on. This would explain the finding that visits that lasted longer had higher scores on biosecurity. Time spent on farms was, however, not significant in the multivariate model, with the number of years into the project staying in the model.

The improved biosecurity in Year 2 was possibly due to an increased knowledge and interest in the topic of biosecurity, as shown by their questionnaire responses and that they then refined their delivery over time leading to a reduction in time and more concise advice. The more time spent delivering the advice and the more experience gained in the field of biosecurity has led to the increases in comfort and capability [19]. It is also suggestive of greater confidence and willingness to make a change on the farm by the PVPs where organisation and communication skills also come into play: communication needs to be concise and effective to transmit the point across to the clients [14]. The importance of listening is also key in this aspect; poor communication skills lead to misinformation, confusion, and errors [20]. It is reasonable to suggest that the use of the biosecurity scoring tool has helped with the effective listening, recording, and advising. The decrease in time that followed, suggests that the PVPs may have honed their communication skills to provide advice more consistently and more efficiently to farmers as the project progressed, as they gained in confidence and improved their skills when discussing biosecurity [21].

As biosecurity visits were becoming time efficient, there was progress made in the overall biosecurity scores that the advisors reported back: a drop in the score represents a better level of biosecurity by increasing the farm’s barriers to disease introduction. The coefficient of variation, that shows the variability of the biosecurity score relative to the mean score, tended to increase at the same time. This was particularly noticeable in PVPs that felt more capable with giving said advice. This suggests that although the overall biosecurity increased, the advice remained variable and therefore specific for the individual project farms, which in turn was reflected in the level of discretion that the PVPs felt when using the scoring tool. This is in line with previous findings, where the ability to provide farm-specific biosecurity advice related with changes in farmer behaviour [22].

The PVPs that reported to ‘Strongly agree’ with an increased interest in biosecurity also reported higher biosecurity scores, suggesting a poorer level of biosecurity. Additionally, their variation of the scores across their farms is higher. These findings could be explained by an increased level of engagement and scrutiny during their visit on the farm, making sure all risks are identified and weighted in the scoring tool. This is echoed by the finding that PVPs reporting to have an increased knowledge on biosecurity also returned higher mean scores on their farms. By scoring the farms more thoroughly, the risks are better identified allowing fuller and therefore better communication between of farmers and PVPs. This allows better knowledge exchange and understanding with the farmer, which has been shown to have influenced farming practices [13,23,24].

Improved biosecurity resulting in reduced presence of diseases is fundamental to producing better welfare and safer consumables for people [2,25]. In this study, the median scores on the questionnaire obtained from the PVPs before they began the study compared to at the end show increases on all aspects of providing a competent advisory service. All but the level of discretion increased significantly. The perceived discretion could be affected by the use of a biosecurity scoring tool in the project. The structured biosecurity tool may have led to more targeted advice towards areas of biosecurity that needed the most improvement, rather than allowing a large level of discretion in their advice. In terms of biosecurity and disease prevention on farms, poor communication could lead to increased production losses, loss of trust in the PVPs and could harm the relationship between farmer and PVP, making them less likely to listen to the PVP’s advice [26]. This compromise between farmers and PVP seemed to be one of the key aspects to improve. The data showed that the PVPs’ perceptions of farmer advice uptake has increased throughout the project, suggesting that the project or the use of the tool may have reduced the compromise between the farmers and their PVPs. The repeated nature of the project and the shared interest of both parties has possibly contributed to this as literature shows how the relationship between PVP and client is fundamental, especially when it comes to farm clients [8].

The increased level of competence through feeling more comfortable, capable and consistent when giving biosecurity advice suggests that the PVPs at the end of the study felt they had improved their abilities. There was also a PVP reported sense of increased uptake of advice by the farmer, which was confirmed by the drop in numerical score on biosecurity. The scores for all categories showed changes, which would suggest that the extra training, advice, and support provided by the project to the PVPs allowed them to develop to be competent in biosecurity out in the field. Of course, many other factors could have an effect here, such as experience, years qualified, and previous training. It is interesting, however, to see that PVPs who perceive themselves to be role models for farmers felt comfortable at the start of the project, as well as in Year 2, but this was not the case in the first year. This could reflect the learning cycle of the conscious-competence model [15,16]. This learning model is used in the development of clinical reasoning. At the first stage a naïve (unconscious incompetent, prior) outlook is proposed, followed by a conscious incompetence (Year 1) and conscious competence (Year 2) and ultimately after more exposure unconscious competence would follow [15,16]. PVPs may be in different phases of their development. This could explain why the effectiveness of utilising PVPs to spread information and educate farmers on biosecurity is well founded but not always necessarily successful [22,27]. This could be alleviated by a problem based practical training of PVPs, to bring them to the unconscious competent phase effectively. This allows PVPs to give high quality biosecurity advice—and charge for it.

Overall, the increase of knowledge and interest in biosecurity of the PVPs show that these individuals will feel better prepared to improve biosecurity on farms in the future. Better communication and education of farmers and their advisors has been shown to have influenced farming practices in the past and there is no reason to believe that this is not also the case when it comes to biosecurity [13,23,24]. Furthermore, from the open questions asked to the PVPs at the end of the study, despite feeling the need to hassle farmers, the overall response was positive, appreciating the opportunity to engage with disease control on their beef suckler farms, as on beef farms in the UK the veterinary involvement is relatively low. On dairy farms, there is a programme in the UK that focusses on the control of Johne’s disease (*Mycobacterium avium* ssp. *paratuberculosis*) that is using trained and certified practicing PVPs to set out control plans as part of their farm assurance [28]. A similar model could be used to improve the biosecurity training for PVPs. An increased veterinary involvement with disease control on farm, could result in increased welfare and production on farms and an improved veterinarian–farmer relationship.

## 5. Conclusions

The reported research shows that there is still room for improvement when it comes to delivering biosecurity advice to farmers by PVPs. The professional relationship has been highlighted as a key factor in the promotion of good biosecurity practice. In addition, the increased competence in PVPs allows for an increased knowledge exchange. The study has shown how exposure to enhanced biosecurity training has led to an increase in competence in the PVPs when providing advice. The increased veterinary competence in disease control on farms followed the conscious-competence learning model. The training on biosecurity, the disease testing and advising on farm, and the use of the biosecurity scoring tool facilitated this professional development.

## Figures and Tables

**Table 1 animals-11-01969-t001:** Median and interquartile range (IQR) of the reported qualifiers associated with giving biosecurity advice to the farmers.

Marker of a Sense of Competence	Baseline Median (IQR)	Year 1 Median (IQR)	Year 2 Median (IQR)	*p*-Value
Comfortable	7.50 * (7.00–8.38)	8.00 (8.00–8.88)	9.00 * (8.25–9.00)	0.022
Capable	7.25 * (7.00–8.00)	8.00 (8.00–8.38)	8.25 * (8.00–9.00)	0.002
Consistent	7.00 (6.25–8.75)	8.75 (8.00–9.00)	9.00 (8.13–9.00)	0.006
Discretion	6.50 (5.25–8.50)	8.00 (6.25–8.88)	8.00 (8.00–8.88)	0.165
Uptake of advice	5.00 (5.00–5.00)	6.00 (5.00–7.00)	6.50 (5.25–7.00)	0.015

Based on eight private veterinarian practitioners’ responses. Reported *p*-value is the result of the Friedman test. *: Significantly different based on post hoc analysis with Wilcoxon rank test (*p* < 0.05) after Bonferroni correction.

## Data Availability

Third Party Data: restrictions apply to the availability of these data.

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
