# Peer review of "Changing Veterinary Attitudes towards Delivering Biosecurity Advice to Beef Farmers"

_animals, 2021, doi:10.3390/ani11071969_

Round 1

Reviewer 1 Report

Abstract

Line 23-24: This sentence does not read well…

“This study looked at the veterinary perception to aspects of their competence to delivering biosecurity advice as part of a longitudinal biosecurity project.”

Suggestion…

This study looked at veterinarians’ perceptions of their own competence to deliver biosecurity advice as part of a longitudinal biosecurity project.

Introduction

In general the sentence structure does not help the reader. Shorter more concise/direct sentences would help bring the reader with you.  

 Line 40: Some areas of farming, such as pig systems, have imposed stringent biosecurity measures…

Suggestion…

Some farming sectors, such as the commercial pig industry, have imposed stringent biosecurity measures…

 Line 42-48: The wording around the cattle industry is pretty superficial and could be linked more directly to biosecurity issues.

Control measures… More involved management… such as?

Beef section is particularly wordy: “More distant approach may take place because...”  More extensive systems are employed 

Line 52: Define abbreviations, DEFRA, APHA (Obvious if UK but not necessarily to wider audience).

Line 57: Missing word in this sentence?

“These professionals are/may be perceived as detached from the issue and not having the farmer’s best interests at heart…”

Line 59: Missing words in this sentence?

“way to increase uptake of biosecurity measures.”

Line 72-74: Have I understood your meaning correctly here?  That farmers may want to do the same as their neighbours (who they talk to) and not necessarily what is best for their individual farm…?

I think you need to work community in somewhere to improve the clarity for the reader.

“A local practicing veterinarian may be faced with farmers who as a community want to adopt a single solution approach to biosecurity issues, which is not a realistic option in many endemic diseases.”

 Line 76: “a key role in providing a greater variety of biosecurity advice, therefore.” Not sure the therefore is needed??

Line 80 - This sentence is hard to follow.   Who is having confidence in the vet’s competence? Is it the vet themself or the farmer? Can you re word to make this clear. Shorter sentences with one idea each would make this easier to follow.

Similarly other sentences in this section are long and awkward to follow.

Materials and methods

Line 100:  five not 5

Please work into methods section whether a high biosecurity score is preferable to a low one or vice versa. Very long sentence in final paragraph could be tightened up to explain what was done more concisely.

Participant perceptions – it’s really quite hard to follow this section. I recommend it is looked at from the perspective of someone trying to repeat the study.  I couldn’t repeat the study from the detail given. I don’t have access to the appendix – can the questions be included in the manuscript to make it clearer what is being discussed?

Applicable to what?  “most applicable” “least applicable” -

 Role model – this term needs to be defined is this the vet as a role model to the farmer, or to other vets, or whether the vet has a role model…?

Please define comfortable, capable, consistent and what you mean by discretion in giving biosecurity advice…  How did you explain to the participating veterinarians what these terms meant in your context to ensure they were all answering the same question? These words could well mean different things to different people. I thought reading Cardwell et al. (2016) might help with this but it doesn’t, your study needs to be completely described to the reader, so they can understand what has been done look at the results and see if they agree with your discussion.

Line 131…missing word?

The vets were additionally asked to rate? their overall project experience?? participation (improved or increased knowledge, interest, involvement, likely role model or likelihood of charging for biosecurity advice) once the study had taken place using answers based on a 5 point Likert scale  (1 = Strongly Disagree to 5 = Strongly Agree).

Suggestion…

At the end of the study participating veterinarians were asked to reflect on their experience of the project and rate themselves (5-point Likert scale; 1 = Strongly Disagree to 5 = Strongly Agree) against the following statements:

  • I have improved or increased knowledge in…
  • I have improved or increased interest in…,
  • involvement,
  • likely role model
  • likelihood of charging for biosecurity advice

If we had the statements, we would know what they were asked about…

Lines 131-141: Long-winded.

The amount of time spent on providing biosecurity advice by the vet to the farmer was also recorded at the different stages of the study. This was further divided into the shortest time taken to give advice, the average time taken to give advice and the longest time taken to give advice.  

This doesn’t need to be said three times in the same sentence it just makes it hard to read…

Throughout the study (Before, Year 1, and Year 2 visits) the length of vet-farmer biosecurity consultations was recorded (in minutes) capturing the minimum/maximum and average time spent delivering advice.  

Line 145: Data not Date?

There is just one sentence in the methods to say free text themes were looked but there is no expansion on how or what was looked for which would help the reader with the results. Was this manual or did you use software?

 Results

Eight, out of how many veterinarians?

What was your response rate?

Did you have any partially complete questionnaires?

 Line 166 – is incomplete or could be shortened. 

Line 170 – what do you mean by ‘level up advice’? Should this be an ‘of’?

This data might be more easily followed if it was presented in a table.

Line 184: This seems to be methodology in the results section. A minimum/maximum range and average consultation time would be much easier for the first-time reader to understand.

First mention of no-occasionally and always-always we need more details of the survey questions and responses to follow the results reporting.

Line 189: Discussion of results in the results section.

Table one: Include N in the legend. (Reader should be able to fully interpret the table without reading the text)

Line 195: ah :0) Re role model: Does the vet perceive themselves as a role model for the farmer with respect to biosecurity…

Line 245: This sentence does not read well.

This study describes the developmental progress made by veterinary participant that give biosecurity advice.

 Line 248: “…training during the project and their and their ability to deliver advice effectively and eloquently…”

  • Some duplicated words in here

Line 255: this section is a bit repetitive of what has just been said.

Line 268: this sentence is awkward with the two ‘the’s.

Line 269: This statement needs to be made much earlier on in the manuscript :0)

Line 271: Can you illustrate for the reader what the term CoV means with respect to the questionnaire before you go onto suggest a reason why it might vary.

Line 274: these terms need to be defined they might /probably do mean different things to different people.

Line 276: can you be more concise here? It would also help to remind the reader of the result being discussed.

Line 283: “Better communication and education in farmers and vets has been shown to have influenced farming practices in the past and we are possibly observing a similar trend”

  • Is in the right word? This statement is a bit fluffy/ informal here... can you make it clearer/stronger?

Line 286: Reference needed to support this statement.

Line 296: Has number reference, and author/date reference.

Line 298: “Farmer uptake was analyzed and shown to increase throughout the project.”

Is this included in the methods section?

Line 298-299:  Illustrate your statement “the literature shows…

Lines 308-310: you need a reference for the conscious-competence model here. You have hyphenated the two words in the summaries ahead of your manuscript.

You could introduce this C-C model ahead of the study (in your introduction) and then you would be able to discuss your results in the light of the model. At the moment the first mention of the C-C model is in the discussion. Shorter sentences would make your argument easier to follow in the discussion.

Lines 329-332: It is very hard to follow the meaning of this very long sentence.

Which model do you mean by "This model"? Yours or the one used for Johnes?

Again shorter sentences would help the meaning. I also think there are some important words missing which makes it easy for the reader to get lost.

Lines 335-337: Try reading this sentence aloud does it say what you think it does? Your meaning is not entirely clear to me...

Line 338: exposure to advanced/ enhanced biosecurity training. Not just biosecurity…

Line 339: Very long-winded can you reword this to get your take-home point across more concisely. Again one topic per sentence (shorter sentences) would be easier for the first-time reader to follow.

Line 341: give an example of how the tool has helped…

Eight seems a small number of respondents to base a study on – as a potential limitation it would be good to see this addressed in the discussion. You may be able to argue it was a 100% response rate for vets involved in the original study (but you haven’t told the reader how many questionnaires were sent out). You may be able to cite others who have used similar sample sizes…?

Reviewer 2 Report

In my opinion is an interesting study with results that despite belong to a survey conducted some years ago, still is worthy to be published as covers a very interesting and relevant topic.

I do only have some comments on the part of materials and methods as several parts where unclear (at least for me).

For example, is not completely clear which is the understanding of the aspects covered in the survey: comfortable, capable and consistent giving advice, the level of discretion used and the uptake of the biosecurity advice. I could not found the questionnaire that is supposed to have been attached in the appendix. Maybe, some of the doubts related to the exact meaning of the aspects covered with the survey could be clarified by reading the questions included in the survey, but as it seems to not have been included in the paper, I cannot evaluate this part.

In addition, I found some sentences confusing and maybe authors could consider to rephrase them:

“Date were first summarised in mean and standard deviation (SD), parametric data, or median and interquartile rage (IQR) for the non-parametric data.”

“Evaluating the impact of the perceptions of the veterinary advisors as well as their time spent doing the biosecurity visit on the veterinarian specific biosecurity score, measured as mean score, the standard deviation (SD) of their scores and the coefficient of variation (COV).”

Is also not clear  why significant levels after the Bonferroni correction have been set to p<0.017 and not the usual p<0.05 and I also have doubts on why treating the participant’s mean, SD and COV biosecurity scores as different variables. If there is an observation per participant and year, why not using the score distribution as the response variable instead of creating three different variables. If authors consider this is a reasonable approach, further details should be provided in relation to the interpretation of SD and COV observed differences. 

Reviewer 3 Report

Dear Author(s),

Your manuscript contains valuable information and is overall interesting to read. The language should be reviewed however, as the language used are often colloquial and still contains some mistakes. Some key points to understanding the methods are missing. These general remarks should be improved, along the following remarks in detail.

Kind regards,

Simple summary:

  • Many different factors: What other factors than the ones discussed here. Add in introduction.
  • their competence (l11): what competence is that? 
  • l14: compared to? 
  • l15: famers => farmers, and had improved instead of has
  • l16-17: why? Could be just as easily more work and discussion in detail at the start, while later only details are left to discuss. That would be consistent with literature? Furthermore, not all vets would develop the same, so the previous explanation seems to be the more logical one to me. 

Introduction:

  • l37-42: understandable but oddly phrased: any practice, 'pig systems'
  • l44: yet other difficulties pop up in the dairy industry: milking, bigger groups, etc.
  • l57-58: I believe the word 'considered' was omitted? If not, the sentence is not correct, nor is the content: professionals can be considered as not caring, but is that necessarily the case?  
  • l63: In particular? Many other, more important topics come to mind more easily: communication, considered lack of results, changing the farmer's habits, etc. 
  • l72-74: This could actually be considered help instead of a challenge. Since one of the measures works, maybe the others are valuable too. 
  • l76: therefore seems to be redundant here? 
  • l77: results are difficult to show in biosecurity, so why would their regular vet be better at showing them? 
  • l80: whose confidence? of the vets themselves, of the farmers in the vets? 
  • l83: although I agree that advice is important, there are many other factors at play. Also, in my opinion words such as 'incredibly' do not belong in a scientific publication. Please refrain from using such words throughout the text. 
  • l85: and so? please rephrase. 
  • l88: compared to? 
  • l91: why beef? this is not explained throughout the text.

M&M

  • l106: how?
  • l107: which aspects? Also, follow on meeting?
  • l110-114: How? did you take in account the vet, how he filled in the survey? What bias? What was done to solve it? Were the 5 previously mentioned diseases used? What diseases were used? A general level of biosecurity? How did the system work? What aspects of biosecurity were used? 
  • l116: current disease? only one disease active on the farm? Also: for can be removed, I believe? 
  • l122: Of should be removed. 
  • l127-128: jumping between different tenses. Also, how often were the interviews done? Once every year? And how? No bias concerning that is described, or how it was solved. 
  • l129: system of 1-10? Please explain.
  • l132-135: How was this checked? 
  • l137: How was this divided? How were the themes identified?
  • l145: Date => Data, I presume. A mean is not the 'summary' of data, please rephrase. parametric data is not part of the sentence, please rephrase. 
  • l149: a p value was not set at 0.05, nor is difference tested, the difference was considered significant if the p value was lower than 0.05. 
  • l151: why was the pvalue set at that level? 
  • l 160: please use passive tense where possible.

Results: 

  • l166: giving advice on?
  • l167: Nothing is mentioned of these scores before. How should they be interpreted? Is 150 good?
  • Why the difference is shortest, longest etc? How did you make a difference? 
  • l169-172: What does this mean? 5 numbers for 5 parameters, but is that overall? for before, year 1, year 2? This is not clear, and values seem to be missing. Furthermore, values seems to be the subject so the verb should be were. Next, a few correlations were identified...
  • How was consistency, capability etc. assessed? Did the vet judge himself, the researchers? 
  • l184: spent
  • l185: Why is the performance measured like that? Why is the amount of time spent a good measurement for the performance?
  • l209: there is no way to comprehend this! 
  • l211: so biosecurity got worse? 
  • l212: Interesting, but would it not be logic that someone who is willing to change, would be more interested in advice and explanations on the topic, causing the visit to be longer? This is not discussed anywhere. 
  • l223: what is the other measure of the biosecurity score? 
  • l224-227: comparing biosecurity scores I presume? This is not clear. 
  • l235: this sentence is incorrect. Verb should be plural, and the next sentence should probably start at l237? Were do vaccin sales come in? 
  • l239: they increased involvement? Unclear, was there increased involvement on farm, did they increase their involvement, did the farmers increase it? 
  • Some "quotes" are in italic, others are not.

Discussion

  • l245: by veterinary participant
  • l248: and their and their. Also, whose ability?
  • l250-263: This is a highly suggestive conclusion. As mentioned before, the overall process in literature and in practice for biosecurity guidance is an initial visit that takes several hours, with several shorter follow-up visits. This seems to fit this description as well, without comfort and confidence. Confidence indeed is bound to grow after gaining experience, but no causality can or should be assumed. 
  • l268: of should be removed. Also progress was made. 
  • l269-270: Ok, so here we learn that a lower score is better? Yet still we have no reference.
  • l273-275: This can be found in several publications, reference? 
  • l276-280: Possible bias? conforming bias? Possible bias is not described anywhere in the manuscript, while the possibilities and impact could be significant. This should be at the very least discussed.
  • l283: but are all risks covered in the tool? It is not described anywhere in the manuscript. 
  • l286: biosecurity against production loss diseases? 
  • Three different ways of references have been found throughout the manuscript
  • l302: sense of increased uptake? Was it quantified? How was it measured actually? 
  • l310: the first year in? 
  • l311-314: Is it that simple? Do all vets learn at the same speed? Did all vets learn and keep using it? This is all based on the same premise as l250-263. 
  • l318: Charging is mentioned a couple times, yet no explanation is given throughout the manuscript. 
  • l321: communication with and education in farmers? Also, education in farmers was hardly mentioned up to now, the manuscript handled training of veterinarians, not farmers. 
  • l325: Was the response really overwhelming? This word does not belong in a scientific publication. 
  • l330: in veterinary? 
  • l331: increase involvement?

Conclusions

  • l335-336: either both or as well as.
  • l338: exposure to biosecurity? 
  • l339: whose involvement?
  • l341: disease testing: this has not been explained throughout the manuscript. 
  • l342: facilitated what? 
  • Throughout the manuscript: Who judged the "competence" of the vets? They themselves, the farmers, the researchers? As the competence was not assessed objectively, it should at least be described as the perceived competence, perceived capability etc. 

References: 

  • Most references handle cattle or biosecurity, none of very few handle biosecurity in cattle. A lot of recent articles can be found on this topic.
